# Could a ring treatment approach be proposed to control *Taenia solium* transmission in a post elimination setting? A pilot study in Zambia

Chiara Trevisan[1,2]*, Kabemba E. Mwape[3], Inge Van Damme[2,4], Ganna Saelens[2], Chishimba Mubanga[5], Mwelwa Chembensofu[6], Maxwell Masuku[3], Seth O'Neal[7], Gideon Zulu[8], Pierre Dorny[9], Sarah Gabriël[2]

**1** Department of Public Health, Institute of Tropical Medicine, Antwerp, Belgium, **2** Laboratory of Foodborne Parasitic Zoonoses, Ghent University, Merelbeke, Belgium, **3** Department of Clinical Studies, School of Veterinary Medicine, University of Zambia, Lusaka, Zambia, **4** Service Foodborne Pathogens, Sciensano, Brussels, Belgium, **5** Ministry of Agriculture, Shibuyunji, Zambia, **6** Department of Paraclinical Studies, School of Veterinary Medicine, University of Zambia, Lusaka, Zambia, **7** School of Public Health, Oregon Health & Science University and Portland State University, Portland, Oregon, United States of America, **8** Ministry of Health, Lusaka, Zambia, **9** Department of Biomedical Sciences, Institute of Tropical Medicine, Antwerp, Belgium

* ctrevisan@itg.be

**Data Availability Statement:** The authors confirm that all data underlying the findings are fully

## Abstract

### Background

Geographically targeted *Taenia solium* ring approaches consisting of treating individuals within a radius of 100-meter of a cysticercosis positive pig have been trialled in Peru. This study explored if a similar approach could be proposed to control *T. solium* transmission in a post elimination setting in Zambia, focussing on community members' willingness to be sampled and treated.

### Methodology and Principal findings

The study was conducted in a community where elimination of active *T. solium* transmission was achieved. All eligible pigs and people were sampled, at 4- to 6-monthly intervals, followed by implementation of the ring treatment approach. This implied that whenever a pig was seropositive for cysticercosis during sampling, every human and pig residing in a radius of 50-meters of the seropositive pig would be treated. The results of the positive human stool samples were used to create the rings, whenever no pigs were positive.

From June 2018 to October 2019, four samplings, followed by ring treatments were conducted. Between 84% and 91% of the willing people provided a stool sample, covering 46% to 59% of the total population living in the study area. Between 78% and 100% of the eligible pigs got sampled. Three ring treatments were based on porcine seropositivity and one on taeniosis results. Two to four rings were opened per sampling. During the ring treatments, between 89% and 100% of the eligible human and pig population living within a ring was treated.

available without restriction. All relevant data are within the paper.

**Funding:** The study received financial support from the Institute of Tropical Medicine, Antwerp (https://www.itg.be) via the Flemish Department of Science, Economy and Innovation (EWI, https://www.ewi-vlaanderen.be/) to S.G. The funders had no role in study design, data collection and analysis, decision to publish, or preparation of the manuscript.

**Competing interests:** The authors have declared that no competing interests exist.

## Conclusions

Participants were willing to participate and get treatment, once the rings were opened. However, the utility of ring treatment approaches in a post elimination setting needs further evaluation, given the lack of highly accurate diagnostic tools for porcine cysticercosis and the challenges in obtaining stool samples. The ring treatment approach adopted should be further improved before recommendations to public health authorities can be given.

## Author summary

In this study, we looked at whether a method used to control the spread of the pork tapeworm, a parasite transmitted between pig and people, in Peru could also work in Zambia. The method involved treating people and pigs living near animals infected with tapewormlarvae.

We conducted the study in a community where the spread of the parasite via infected meat had been successfully stopped. Every few months, we sampled pigs and people in the area and treated those who were found to be infected or at risk. If a pig tested positive for the parasite, everyone within 50 meters of that pig was treated.

Between June 2018 and October 2019, we did four rounds of sampling and treatment. Most people were willing to provide stool samples, and a large percentage of pigs were sampled as well. During treatment, most eligible people and pigs within the affected area received treatment.

The study found that people were willing to participate and get treated once the treatment rings were set up. However, there are challenges with accurately diagnosing the parasite in pigs and getting stool samples from people. Before recommending this method to health authorities, further improvements are needed.

## Introduction

Over the last decade, highly efficient *Taenia solium* intervention tools have been developed and evidence-based control/elimination strategies have been trialled [1]. Addressing the parasite from a One Health perspective, targeting both humans and pigs, has shown to be effective in eliminating *T. solium* transmission in endemic countries such as Zambia [2] and Peru [3]. These achievements required repeated rounds of human and pig mass drug administrations (MDA) combined with pig vaccinations and extensive community education. These trials were guided by transmission models that assisted in determining which combination of control tools to integrate, and in predicting infection reduction and elimination of transmission, however not taking into account the longterm sustainability and feasiblility of the approach [4,5,6].

Ongoing health education, treatment of tapeworm carriers and of incoming pigs, combined with meat inspection, have been suggested as other potential strategies [7]. In addition to these, targeted "ring approaches", such as ring screening and ring treatment were designed and trialled in Peru, showing to be effective and feasible *T. solium* transmission control options [8,9]. In Peru, ring screening consisted of active surveillance for heavily infected pigs by performing tongue palpation on all pigs in the village. If a pig was found positive, a 100-meter radius ring around any tongue-positive pig was opened and all residents within the ring were

tested for taeniosis. Targeted treatment was then given to people testing positive. Ring treatment consisted of treating all individuals residing within a 100-meter radius around any tongue-positive pig, without prior knowledge of the taeniosis infection status. In both approaches, infected pigs were removed from the village. In case a farmer disagreed to remove the infected pig, pig treatment with oxfendazole (single dose, 30 mg/kg) was offered [8,9]. These approaches were particularly effective as they took advantage of the spatial clustering nature of the disease and foresaw targeted treatment in areas where the risk was concentrated. Despite Peruvian promising results, it remains unknown whether these approaches, or a combination thereof, would also be acceptable in endemic regions with different socio-cultural and geographical characteristics such as in sub-Saharan Africa.

Our previous research indicated that a 2-year intergrated One Health approach could eliminate *T. solium* transmission in Zambia [2], though model predictions have shown that additional long-term efforts need to be made to maintatin and sustain control/elimination, prevent re-infection from adjacent areas and avoid wasting of resources [6]. Therefore, this study aimed to 1) explore the feasibility of implementing a modified ring treatment strategy in a sub-Saharan African setting, indpired by a similar approach in Peru, focussing on the willingness of the local community residents and pig farmers to participate in such an approach, expressed by willingness to be sampled and to be treated, and 2) to provide future recommendations on sustainable control of the parasite in a post elimination setting, based on knowledge gained in the targeted context.

## Methods

### Ethics statement

Approval for the study was granted by the University of Zambia Biomedical Research and Ethics Committee (004-09-15, covering both human and animal ethical clearance), the Ethical Committee of the University of Antwerp, Belgium (B300201628043, EC UZA16/8/73), and the Institutional Review Board of the Institute of Tropical Medicine (1023/14).

Participants were informed about the procedures, the sampling methods and the risks and benefits of participating in the study. A written informed consent was obtained from all the particpiants willing to take part of the sudy. Participants younger than 18 years were asked for verbal assent as well as a written consent from their parent or guardian. Illiterate participants were asked for oral assent, with written consent signed by an impartial witness. Pig owners were requested to give permission before each pig sampling/treatment round. Everybody was informed that participation was voluntary and that participants could withdraw from the study at any time point if they preferred to do so, without any consequence.

### Study area

The study was part of CYSTISTOP, a large-scale community-based project where different interventions were trialled to achieve *T. solium* control/elimination in highly endemic communities of the Eastern Province of Zambia [2]. This study was conducted in the Nyembe neighbourhood, in the Katete District, the study arm where active *T. solium* transmission in pigs was ieliminated. This achievement was possible after two years of intensive 4-monthly interventions, addressing the pig and human hosts, with pig vaccination and human and pig MDA treatment, including health education [2]. The Nyembe neighbourhood consisted of 1,084 people and 184 pigs at baseline, and 1,135 people and 72 pigs at the post elimination intervention stage, distributed across 226 households in eight villages.

## Study design

The ring strategy was implemented during the monitoring and surveillance phase of the project, where follow up samplings were conducted on humans and pigs at 4- to 6-monthly intervals. The ring strategy implied that whenever a pig was found seropositive for cysticercosis during a follow up sampling, every human and pig residing in a range of 50-meter radius of the infected animal would be treated with niclosamide or praziquantel, and oxfendazole, respectively (see further details in section 2.5). In case no pig was found positive, the results of the human samplings were used to create the 50-meter rings around a household with a taeniosis positive family member.

As the villages and the households within the villages were geographically close to each other, a 50-meter radius was chosen to avoid a full village MDA from the moment positive pigs would be identified in a few different households.

## Human and pig sampling procedure

After household registration, stool pots were distributed to consenting human participants in the study area. Eligibility included: i) being at least 5 years of age, ii) willing and able to participate in all aspects of the study, including sampling and treatment, iii) willing and able to provide written informed consent, iv) not being seriously ill and v) living in the study area. Human stool collection was conducted at village level by four different teams, until each village was covered.

All pigs living in the villages were registered. To be eligible for sampling, pigs had to be: i) at least two months of age, ii) neither lactating nor pregnant, and iii) not showing signs of illness. Blood samples from eligible pigs were taken by qualified veterinary personnel after obtaining permission from the pigs' owner. In addition, tongue palpation was performed to assess if cysticerci were visible or palpable on the pig's tongue.

## Laboratory procedures

The samples were initially processed at the field laboratory in the study area prior to transportation to the University of Zambia, School of Veterinary Medicine (Lusaka, Zambia) for further analysis.

Subsequently, the human stool samples were analysed with the copro-antigen enzyme-linked immunosorbent assay (ELISA) following the method described by [10] and modified by [11]. The pig serum samples were analysed with the B158/B60 serum-antigen ELISA following the method described by [12]. The antigen ELISA was preferred over the antibody ELISA to indeitfy pigs with viable cysticerci, as the antibodies remain positive long after treatment.

## Treatment

Treatment was given to humans and pigs residing within a radius of 50-meters around a porcine cysticercosis seropositive pig or, in case no pig was found positive, around a taeniosis coproantigen positive human. Oral treatment with niclosamide (2 g single dose for adults and 1 g for children 2 to 6 years old) or praziquantel (10 mg/kg) was given to humans. Pigs were orally treated with a single dose of oxfendazole (30 mg/kg).

## Data management and analysis

Demographic, sampling and treatment data was entered on tablets using EpiCollect https://five.epicollect.net (2022 Centre for Genomic Pathogen Surveillance, v4.2.0). Data consistency checks, data cleaning and analyses were done with R version 1.3 [13]. To identify individuals

**Table 1. Human follow up samplings coverage.**

| Follow up sampling | N° of HHs | Total human population | Pots distributed | | Human stool samples received | | |
|---|---|---|---|---|---|---|---|
| | | | N° | % of THP | N° | % of THP | % of pots distributed |
| FS 1 (Jun'18) | 225 | 1086 | 549 | 51 | 499 | 46 | 91 |
| FS 2 (Oct'18) | 216 | 1071 | 665 | 62 | 587 | 55 | 84 |
| FS 3 (Apr'19) | 232 | 1062 | 696 | 66 | 622 | 59 | 89 |
| FS 4 (Oct'19) | 224 | 1037 | 606 | 58 | 522 | 50 | 86 |

HH: Household; FS: Follow-up sampling; THP: Total human population (including non eligible individuals); N°: Number

residing within a radius of 50-meters around an infected pig or human, all households were identified based on their global positioning system (GPS) coordinates collected during the baseline sampling in 2015 [2]. Outcomes used to assess willingness were human and pig sampling and treatment coverages.

## Results

### Follow up samplings

A total of four follow up samplings were performed from June 2018 to October 2019. Due to the Covid-19 pandemic, a fifth planned follow up sampling in April 2020 had to be cancelled. Tables 1 and 2 show the coverage for the human and pig follow up samplings, respectively.

During the 4 follow up samplings, between 46% and 59% of the total human population in the study area was sampled, collecting and analysing between 499 and 622 stool samples per follow up sampling. Out of the total stool pots distributed, between 84% and 91% of the people provided a sample.

During the follow up sampling, the eligible pig population ranged between 38 (69% of total pig population (TPP)) to 54 pigs (52% of TPP). The follow up sampling covered between 78% and 100% of the eligible pig population.

### Ring treatments

Three out of four ring treatment rounds were based on the results of the pig samplings (serum positive pigs) and one ring treatment was based on the results of the human stool sample analysis as no pig was found infected. In FS 1, four out of 49 sampled pigs were found positive. Three seropositive pigs were from the same household, hence two rings were opened. In FS 2,

**Table 2. Pig follow up sampling coverage.**

| Follow up sampling | N° of HHs | Total pig population | Eligible pig population | | Pig serum samples received | | |
|---|---|---|---|---|---|---|---|
| | | | N° | % of TPP | N° | %of TPP | % of EP |
| FS 1 (Jun'18) | 12 | 60 | 53 | 88 | 49 | 82 | 92 |
| FS 2 (Oct'18) | 11 | 55 | 38 | 69 | 34 | 64 | 92 |
| FS 3 (Apr'19) | 14 | 103 | 54 | 52 | 43 | 41 | 78 |
| FS 4 (Oct'19) | 13 | 86 | 47 | 55 | 47 | 55 | 100 |

HH: Household; FS: Follow up sampling; TPP: Total pig population (including non eligible pigs); EP: Eligible pig population; N°: Number

**Table 3. Human ring treatment coverage.**

| Ring treatment | N° of rings | N° of HHs in rings | Total human population in rings | Eligible human population in rings | | Humans treated in rings | | |
|---|---|---|---|---|---|---|---|---|
| | | | | N° | % total | N° | % total | % eligible |
| **RT 1** (Aug'18) | 2 | 10 | 51 | 47 | 92 | 47 | 92 | 100 |
| **RT 2** (Nov'18) | 4 | 28 | 119 | 105 | 88 | 105 | 88 | 100 |
| **RT 3** (Jul'19) | 3 | 10 | 41 | 38 | 93 | 38 | 93 | 100 |
| **RT4** (Dec'19) | 3 | 4 | 20 | 19 | 95 | 17 | 85 | 89 |

RT: Ring treatment; HH: Household, N°: Number

no pigs were found positive for porcine cysticercosis. Four 50-meters rings were opened based on results of positive stool samples from four people, residing in four different households. In FS 3, three rings were opened as three households had one seropositive pig each, out of 43 pigs sampled. Based on tongue palpation, only one ring would have been opened as only one pig was found to be tongue palpation positive during FS 3. This pig was also seropositive.

In FS 4, six out of 47 pigs were seropositive for porcine cysticercosis during the follow up sampling, residing in three different households, resulting in three rings.

All eligible pigs and humans residing in households within 50-meters of a positive household were offered treatment. Tables 3 and 4 show the anthelmintic treatment coverage for people and pigs, respectively. Nearly 100% of all eligible people and pigs in the rings were treated during the follow up ring treatments.

Over the four ring treatments, one household had a seropositive pig in FS 1 and FS 4; thus the people and pigs in the households around the positive pig were involved in ring treatments twice.

## Discussion

In this study, we explored for the first time if a modified ring treatment approach could be proposed in a sub-Saharan setting. Our primary outcome was willingness of the local communities to participate, indicated by readiness to undergo sampling and treatment.

**Table 4. Pig ring treatment coverage.**

| Ring treatment | N° of rings | N° of pig keeping HHs in rings | Total pig population in rings | Eligible pig population in rings | | Pigs treated in rings | | |
|---|---|---|---|---|---|---|---|---|
| | | | | N° | % total | N° | % total | % eligible |
| **RT 1** (Aug'18) | 2 | 4 | 24 | 23 | 96 | 23 | 96 | 100 |
| **RT 2** (Nov'18) | 4 | 0 | 0 | 0 | 0 | 0 | NA | NA |
| **RT 3** (Jul'19) | 3 | 5 | 27 | 27 | 100 | 27 | 100 | 100 |
| **RT4** (Dec'19) | 3 | 3 | 24 | 24 | 100 | 24 | 100 | 100 |

RT: Ring treatment; HH: Household, N°: Number

Three out of four ring treatment rounds were conducted based on results of pigs seropositive for porcine cysticercosis and one based on taeniosis results, treating in total 207 people and 74 pigs. Identifying infected pigs and people needing targeted treatment led to a high compliance, with nearly 100% of all people in the rings accepting treatment during the follow up ring treatments. This is comparable with a study in Peru, where participation remained high over several intervention rounds when a ring screening was applied [8]. The high compliance might be explained by the fact that household members were approached individually, informed about the presence of an infected pig or person, leading to a possibly better understanding of the health threat and thus accepting treatment. Also the perception of risk might have increased using this approach. Compared to mass drug administration intervention, a ring approach is more targeted in its nature, which might also add motivation to participate, as was also shown in the study conducted in Peru [8]. This pilot study was conducted within a community that had undergone intensive interventions over the preceding two years as part of a large disease elimination intervention study [2]. The trust and familiarity established with the project team, coupled with the positive perceptions and high levels of acceptability reported in a previous study conducted in the community [14] may have positively influenced the willingness to participate, resulting in high coverage and compliance rates. This highlights also the importance of community sensitizations, trust relations and community participation when interventions are carried out.

In the study conducted in Peru, rings were opened based on positive tongue palpation results to identify pigs heavily infected with the parasite [9]. In our study, we applied tongue palpation in addition to serology. Over the four follow up samplings, only one pig was diagnosed using tongue palpation. Due to the low test sensitivity of the method, 8 ring treatments would have been missed [12]. Therefore light infections might have been missed also in Peru, reducing the effect of the control program on tapeworm carriers and porcine cysticercosis positive pigs [8]. In contrast to the study in Peru, our study was carried out in a post-elimination setting, where infection levels were zero or close to zero. The choice of adding serology (B158/B60 serum Ag ELISA) with a higher sensitivity than tongue palpation, but a lower diagnostic specificity, might have led to false positive pigs, leading to an overtreatment of the pigs and people living in proximity to the household with the test positive pig [15]. Overall, when considering willingness to participate, it is important to acknowledge that blood sampling is more invasive than tongue palpation. As a result, some pig owners might refuse to participate, leading to missed detection of infections, hence it is important to establish a relationship of trust before any intervention [14]. In areas with low transmission rates or when control gains using tongue palpation have plateaued, the implementation of highly sensitive methods may become necessary [16,17]. However, according to Grass (2020) [17], model predictions (using soil transmitted helminths as an example) revealed that in low prevalence settings, test specificity plays a more important role compared to test sensitivity in determining when an MDA should be conducted or not. Hence, the high test specificity of tongue palpation should be considered, and a combination of tests could be applied. Choices also need to be made based on the objective and if disease control or elimination should be achieved [18]. One control approach to trial in future would be to use community participation and focus on pig backyard slaughtering and reporting of infection from pig traders. This would imply that whenever a pig is found or reported infected, the household members and all pigs living within a certain radius would be treated. Following a One Health approach, reporting would be done from the veterinary sector, while treatment could be organized by both the human and veterinary sector, leading to shared responsibility and resource use. With this approach, sample collection and analyses using expensive lab techniques would not be needed, saving time and resources.

In our study, the 50-meter radius selected for the ring treatment was also based on the rationale that activities that contribute to *T. solium* transmission, such as open defecation and free roaming of pigs are concentrated close to the households. During the same time as the ring treatments, we also conducted a study to monitor free roaming pig patterns in the study area. Based on results of the study we concluded that nearly all pigs spend most of their time within 50- or 100-meters [19]. While a 100-meters ring would have probably covered a larger source of possible infections, in this study setting where the villages and the households are close to each other, the selection of a 100-meters ring would have approached a full-scale MDA intervention in some of the villages, highlighting the importance of setting and area dependence.

In our study, one pig keeping household had an infected pig during two follow up samplings, therefore people and pigs residing in the households around the positive pig were involved in multiple rings. Repeated infection of pigs in a household might be explained by either new infected pigs that are purchased from neighbouring areas, a false positive result using the Ag-ELISA, failed taeniosis treatment and thus continued environmental contamination followed by re-infection, re-infection through contact with *Taenia* spp. egg infected environment by a visiting tapeworm carrier or residual infection in the environment [20]. Another possible reason is the fact that in the area, boars and sows were sometimes lent to other pig farmers of neighbouring areas for mating, leading to possible re-infection. Additionally, the possibility of human re-infection cannot be discounted, even in a post elimination setting, as behaviour change is not always guaranteed. Furthermore, a fundamental limitation of the ring approach is its inability to achieve 100% effectiveness, given that individuals and pigs may venture outside the designated 50-meter rings. Moreover, only 50% of the human population was tested, meaning that cases within the other 50% could not be identified or included in the ring treatments and rings were mainly based on pig outcomes.

In our study, pigs within the 50-meter radius were all treated. In the study in Peru infected pigs were removed from the study area, and treatment was only given in case of disagreement. This approach might also reinforce the removal of infected sources, especially as oxfendazole treatment is not efficacious on brain cysticerci [21]. The latter however might not be an issue in Zambia, as the local cooking practices foresee boiling the head with the brain, reducing the risk of infection.

This study had some limitations. Because of the design of the project, the ring strategy was trialled in the study neighbourhood where elimination of *T. solium* active transmission was already achieved. This meant that in the study area, the team was already well known, the study population had been sensitized before and was therefore more aware of the risks compared to a naïve population [15], possibly impacting positively on the coverage results. During the follow up sampling, participants were sensitized and invited to come for sampling. One of the eligibility criteria for the follow up sampling was willingness to participate, making it difficult to get an accurate overview of who was eligible for sampling and treatment or not. Finally also the proportion of eligible pigs duing certain time points might be problematic, creating a possible limitation in using the ring treatment approach.

## Conclusions

Village residents and pig owners were willing to get treatment once the rings were opened. These results suggest that ring treatment approaches could be considered for control of *T. solium* in this region of Zambia, particulary in areas with ongoing endemic transmission. However, the utility of ring treatment approaches in a post elimination setting needs further evaluation, given the lack of highly accurate diagnostic tools for porcine cysticercosis. Furthermore, as obtaining stool samples from the total eligible population remains difficult, new

strategies to open the rings need to be identified. The results of this pilot study suggest that the ring treatment approach is an acceptable strategy if residents and pig owners' participation is considered, but need to be further improved before recommendations to Zambian public health authorities can be given.

## Acknowledgments

The authors would like to acknowledge all the willing participants and pig owners, the community residents and staff involved in the project. The authors would also like to thank Ana Lucia Fajardo Castaneda, Anke Van Hul, Mwelwa Chembensofu, Sandra Vangeenberghe, Sophie De Bock, Stephen Vandenmeersch and Victor Vaernewyck for the help with the data entry.

## Author Contributions

**Conceptualization:** Chiara Trevisan, Kabemba E. Mwape, Inge Van Damme, Seth O'Neal, Sarah Gabriël.

**Data curation:** Chiara Trevisan, Inge Van Damme, Ganna Saelens, Chishimba Mubanga, Mwelwa Chembensofu, Maxwell Masuku, Gideon Zulu, Sarah Gabriël.

**Formal analysis:** Chiara Trevisan, Inge Van Damme.

**Funding acquisition:** Kabemba E. Mwape, Pierre Dorny, Sarah Gabriël.

**Investigation:** Kabemba E. Mwape, Inge Van Damme, Gideon Zulu, Sarah Gabriël.

**Methodology:** Chiara Trevisan, Kabemba E. Mwape, Inge Van Damme, Ganna Saelens, Chishimba Mubanga, Seth O'Neal, Pierre Dorny, Sarah Gabriël.

**Project administration:** Chiara Trevisan, Kabemba E. Mwape, Sarah Gabriël.

**Resources:** Kabemba E. Mwape.

**Supervision:** Chiara Trevisan, Kabemba E. Mwape, Pierre Dorny, Sarah Gabriël.

**Validation:** Chiara Trevisan.

**Visualization:** Inge Van Damme.

**Writing – original draft:** Chiara Trevisan.

**Writing – review & editing:** Chiara Trevisan, Kabemba E. Mwape, Inge Van Damme, Ganna Saelens, Chishimba Mubanga, Mwelwa Chembensofu, Maxwell Masuku, Seth O'Neal, Gideon Zulu, Pierre Dorny, Sarah Gabriël.

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
