## [Decision Letter · Decision Letter 0]

24 Jun 2024

Dear Dr. Trevisan,

Thank you very much for submitting your manuscript "Could a ring treatment approach be proposed to control Taenia solium transmission in a post elimination setting? A pilot study in Zambia" for consideration at PLOS Neglected Tropical Diseases. As with all papers reviewed by the journal, your manuscript was reviewed by members of the editorial board and by several independent reviewers. In light of the reviews (below this email), if you decide to submit a revised manuscript, we would re-consider a significantly-revised version that takes into account the reviewers' comments. 

The reviewers have expressed particular concerns about the population in which the study was conducted, one which has participated in a mass treatment program for elimination of cysticercosis, because of the consequent impact on interpretation of and ability to generalise the results. If you decide to submit a revised manuscript, we expect these concerns and the other issues raised by reviewer 2 to be addressed.

We cannot make any decision about publication until we have seen the revised manuscript and your response to the reviewers' comments. Your revised manuscript is also likely to be sent to reviewers for further evaluation.

Sincerely,

Siddhartha Mahanty, M.B.B.S., M.P.H

Academic Editor

Eva Clark

Section Editor

Reviewer's Responses to Questions

**Key Review Criteria Required for Acceptance?**

**Methods**

-Are the objectives of the study clearly articulated with a clear testable hypothesis stated?

-Is the study design appropriate to address the stated objectives?

-Is the population clearly described and appropriate for the hypothesis being tested?

-Is the sample size sufficient to ensure adequate power to address the hypothesis being tested?

-Were correct statistical analysis used to support conclusions?

-Are there concerns about ethical or regulatory requirements being met?

Reviewer #1: (No Response)

Reviewer #2: (No Response)

**Results**

-Does the analysis presented match the analysis plan?

-Are the results clearly and completely presented?

-Are the figures (Tables, Images) of sufficient quality for clarity?

Reviewer #1: (No Response)

Reviewer #2: (No Response)

**Conclusions**

-Are the conclusions supported by the data presented?

-Are the limitations of analysis clearly described?

-Do the authors discuss how these data can be helpful to advance our understanding of the topic under study?

-Is public health relevance addressed?

Reviewer #1: (No Response)

Reviewer #2: (No Response)

**Editorial and Data Presentation Modifications?**

Reviewer #1: (No Response)

Reviewer #2: (No Response)

**Summary and General Comments**

Reviewer #1: The manuscript PNTD-D-24-00616 is well written and clear in its statements. I do not understand why the ring elimination method was used to evaluate areas in Zambia that have been already controlled for Taenia solium cysticercosis and taeniasis, the authors have to explain the reasoning since the approaches used are expensive and time consuming.

Reviewer #2: (No Response)

PLOS authors have the option to publish the peer review history of their article (what does this mean?). If published, this will include your full peer review and any attached files.

Reviewer #1: Yes: Ana Flisser

Reviewer #2: No
---

## [Editor Report · Decision Letter 1]

25 Jul 2024

Dear Dr. Trevisan,

We are pleased to inform you that your manuscript 'Could a ring treatment approach be proposed to control Taenia solium transmission in a post elimination setting? A pilot study in Zambia' has been provisionally accepted for publication in PLOS Neglected Tropical Diseases.

Best regards,

Jong-Yil Chai

Section Editor

Jong-Yil Chai

Section Editor

The revised manuscript appears to be acceptable by PLoS NTD.

---

## [Editor Report · Acceptance letter]

4 Aug 2024

Dear Dr. Trevisan,

We are delighted to inform you that your manuscript, "Could a ring treatment approach be proposed to control Taenia solium transmission in a post elimination setting? A pilot study in Zambia," has been formally accepted for publication in PLOS Neglected Tropical Diseases.

Best regards,

Shaden Kamhawi

co-Editor-in-Chief

Paul Brindley

co-Editor-in-Chief
